# Fisher-aware Quantization for DETR Detectors with Critical-category Objectives

Huanrui Yang [*1]   Yafeng Huang [*2]   Zhen Dong [1]   Denis A Gudovskiy [3]   Tomoyuki Okuno [3]   Yohei Nakata [3]
Yuan Du [2]   Kurt Keutzer [1]   Shanghang Zhang [4]

## Abstract

The impact of quantization on the overall performance of deep learning models is a well-studied problem. However, understanding and mitigating its effects on a more fine-grained level is still lacking, especially for harder tasks such as object detection with both classification and regression objectives. This work defines the performance for a subset of task-critical categories i.e. the critical-category performance, as a crucial yet largely overlooked fine-grained objective for detection tasks. We analyze the impact of quantization at the category-level granularity, and propose methods to improve performance for the critical categories. Specifically, we find that certain critical categories have a higher sensitivity to quantization, and are prone to overfitting after quantization-aware training (QAT). To explain this, we provide theoretical and empirical links between their performance gaps and the corresponding loss landscapes with the Fisher information framework. Using this evidence, we apply a Fisher-aware mixed-precision quantization scheme, and a Fisher-trace regularization for the QAT on the critical-category loss landscape. The proposed methods improve critical-category metrics of the quantized transformer-based DETR detectors. They are even more significant in case of larger models and higher number of classes where the overfitting becomes more severe. For example, our methods lead to 10.4% and 14.5% mAP gains for, correspondingly, 4-bit DETR-R50 and Deformable DETR on the most impacted critical classes in the COCO Panoptic dataset.

[*]Equal contribution   [1]University of California, Berkeley [2]Nanjing University [3]Panasonic Holdings Corporation [4]School of Computer Science, Peking University. Correspondence to: Huanrui Yang <huanrui@berkeley.edu>, Shanghang Zhang <shanghang@pku.edu.cn>.

Accepted to the Workshop on Advancing Neural Network Training at International Conference on Machine Learning (WANT@ICML 2024).

## 1. Introduction

Object detection is a challenging core application in computer vision, which is crucial for practical tasks such as autonomous driving. Recent DEtection TRansformer (DETR) model (Carion et al., 2020) and its variants achieve state-of-the-art results on multiple detection benchmarks (Liu et al., 2022). However, their performance comes at the cost of large model sizes and slow inference. Then, quantization (Choi et al., 2018; Dong et al., 2020; 2019; Polino et al., 2018; Yang et al., 2021) is typically applied to reduce the memory footprint and inference latency time on cloud and edge devices (Horowitz, 2014). Inevitably, the perturbation of weights and activations introduced by the quantization process degrades the performance of floating-point models. Previous research on quantization (Dong et al., 2019; Yang et al., 2021; Xiao et al., 2023) mainly focuses on a *trade-off between the model size and the overall performance* (e.g., average accuracy for classification and mean average precision (mAP) for detection).

However, a *fine-grained performance objectives* are often more important than the overall performance in the real world (Barocas et al., 2019; Tran et al., 2022). Suppose an autonomous vehicle is processing a scene containing people, vehicles, trees, light poles and buildings as illustrated in Figure 1 (left)[1]. Some non-critical objects (light poles, trees, and buildings) only need to be localized to avoid collision, yet misclassification within this group of categories is not as critical if they are all considered as "other obstacles". On the other hand, *critical classes* such as a person or vehicle require both accurate classification and localization for a safe operation. The overall performance cannot distinguish between an error within non-critical categories vs. a critical object error. In other words, it is missing granularity to represent the true task-critical objectives of real-world applications. Yet to the best of our knowledge, for both post-training quantization (PTQ) and quantization-aware training (QAT), the analysis of the impact on such task-critical fine-grained objectives of object detection models is largely overlooked.

[1]Street scene photo in Figure 1 credits to Google Street View.

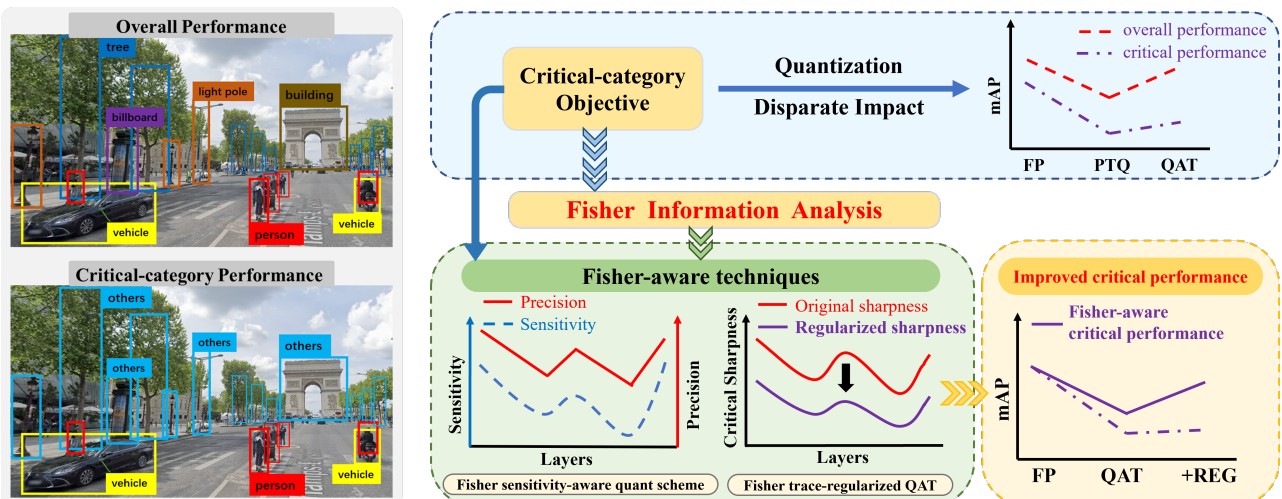

Figure 1: **Overview.** We investigate a practical setting with task-dependent critical-category objectives in Section 3.2. We empirically observe disparate effects of quantization on the critical-category performance in Section 3.3, where post-training quantization (PTQ) and quantization-aware training (QAT) lead to performance gaps for critical categories w.r.t. to a floating-point (FP) model. We theoretically analyze such gaps using Fisher information framework and propose a Fisher-aware mixed-precision quantization scheme with regularization in Section 4 to overcome these gaps for DETR models.

In this paper, we follow this practical yet neglected setting to formulate a set of task-critical objectives for DETR-based object detection models, accomplish a fine-grained quantization impact analysis, and propose techniques for improvements of the corresponding objectives. Specifically, we disentangle classification and localization objectives to define a fine-grained *critical-category performance* with non-critical label transformation, as shown in the updated bounding boxes in Figure 1 (left). With this formulation, we provide a comprehensive analysis of the impact of quantization on the critical-category performance of DETR model. As illustrated in Figure 1 (right), we find that quantization has a disparate effect on the category-wise performance, where some groups of classes are more sensitive to quantization with up to 1.7% additional mAP drop for the public DETR object detector with ResNet-50 backbone. While QAT typically improves the overall performance, it can further increase performance gaps for the defined task-critical categories. We provide both theoretical and empirical analysis of such quantization effects using the loss surface landscape of the critical objectives by applying the Fisher information framework (Perronnin & Dance, 2007).

Based on this analysis, we propose two novel techniques: Fisher-aware mixed-precision quantization scheme and Fisher-trace regularization. Both techniques optimize the landscape of critical objectives and, therefore, reduce overfitting and improve critical-category performance. Our experiments show consistent critical-category performance improvements for DETR object detectors with various backbones and architecture variants. The contributions of this paper are summarized as follows:

- We formulate critical-category objectives for object detection and observe disparate effects of quantization on the performance of task-critical objectives.

- We provide analytical explanations of such quantization effects for DETR-based models using a theoretical link to the Fisher information matrix.

- Our Fisher-aware mixed-precision quantization scheme incorporates the sensitivity of critical-category objectives and increases their detection metrics.

- Our Fisher-trace regularization further improves the loss landscape during quantization-aware training and the corresponding critical-category results.

## 2. Related Work

**Object detection.** Object detection is a core task for visual scene understanding. Conventional object detectors rely on a bounding box proposals (Girshick, 2015), fixed-grid anchors (Redmon et al., 2016) or window centers (Tian et al., 2019). However, the performance of these methods is largely affected by bounding box priors and post-processing steps (Carion et al., 2020). The transformer-based DETR (Carion et al., 2020) provides a fully end-to-end detection pipeline without surrogate tasks. Follow-up research further enhances DETR by introducing a deformable attention (Zhu et al., 2021), query denoising (Li et al., 2022), and learnable dynamic anchors as queries (Liu et al., 2022).

With the growing popularity of DETR-based architecture, we believe that understanding of quantization impact on DETR performance is an important topic, especially at the fine-grained level. Common object detection benchmarks evaluate fine-grained performance metrics that depend on the object size (Lin et al., 2014) or occlusion status (Geiger et al., 2012). However in practical applications, object type, i.e. its category, is often more important than the object size. This motivates us to further investigate detectors with critical-category objectives.

**Efficiency-performance tradeoff.** Multiple methods have been proposed to compress deep neural network (DNN) models, including pruning (Han et al., 2015; Wen et al., 2016; Yang et al., 2020b; 2023), quantization (Polino et al., 2018; Dong et al., 2020; Yang et al., 2021; Guo et al., 2022), factorization (Wen et al., 2017; Ding et al., 2019; Yang et al., 2020a) and neural architecture search (Wu et al., 2019; Cai et al., 2020). In this work, we explore the impact of quantization that is widely supported by the hardware (Horowitz, 2014) and can be almost universally applied to DNN model compression in architecture-agnostic fashion.

In previous work, average post-compression performance metrics are the key focus, but such overall performance hides the important fine-grained metrics e.g., the results for certain groups of categories. Recent works (Tran et al., 2022; Good et al., 2022) analyze the disparate impact of pruning on classification accuracy, which leads to the fairness concerns (Barocas et al., 2019). Our work extends this direction and investigates quantization effects of object detection at the critical-category performance granularity.

**Second-order information in deep learning.** Unlike conventional optimization with the first-order gradients, recent research shows that the use of second-order information increases generalization and robustness of DNN models. Sharpness-aware minimization (Foret et al., 2021) links a loss landscape sharpness with the model ability to generalize. The latter can be improved using a regularized loss with the Hessian eigenvalues (Yang et al., 2022) computed w.r.t. parameter vector. Hessian eigenvalues are also used as importance estimates to guide the precision selection in mixed-precision quantization (Dong et al., 2019; 2020; Yao et al., 2021). Given the difficulty of exact Hessian computation, Fisher information matrix is proposed as an approximation of the importance in pruning (Kwon et al., 2022). In this work, we link the quantization impact on the critical objectives with the second-order Fisher information.

# 3. Quantization Effects on Critical-Category Performance

In this section, we introduce conventional training objectives for DETR in Section 3.1; formulate our critical-category ob-

jectives in Section 3.2; and empirically analyze quantization effects on such critical-category performance in Section 3.3.

## 3.1. Conventional DETR Training

Let $x$ be an input image from a dataset $\mathcal{D}$. Then, the DETR-type model $f_{\boldsymbol{\theta}}(\cdot)$ with weights $\boldsymbol{\theta}$ outputs a fixed-size set of $N = K|\mathcal{D}|$ predictions $\hat{y}_i = \{(\hat{\boldsymbol{p}}_i, \hat{\boldsymbol{b}}_i)\}_{i=1...N}$, where $K$ is the model-dependent number of detections in each image, $\hat{\boldsymbol{p}}_i$ is the vector of classification logits and $\hat{\boldsymbol{b}}_i$ is the vector of bounding box coordinates. The former $\hat{\boldsymbol{p}}_i \in \mathbb{R}^{C+1}$ contains logits for $C$ classes and an empty-box class ($\varnothing$). The predicted bounding box $\hat{\boldsymbol{b}}_i \in \mathbb{R}^4$ consists of 4 scalars that define the center coordinates as well as the height and the width relative to the image size.

During the training, annotation is provided for each image in $\mathcal{D}$ as a set of ground truth objects $y_i = \{(\boldsymbol{c}_i, \boldsymbol{b}_i)\}$, where $\boldsymbol{c}_i$ is the one-hot vector with target class label and $\boldsymbol{b}_i$ defines the bounding box. A Hungarian matching process is performed to find the closest one-to-one matching between ground truths and predictions including those with "no object" $\varnothing$ predictions. The training loss is computed between each pair of matched boxes, which is defined as a linear combination of a classification loss $\mathcal{L}_{cls}(\hat{\boldsymbol{p}}_i, \boldsymbol{c}_i)$ for all predictions, and a box loss $\mathcal{L}_{box}(\hat{\boldsymbol{b}}_i, \boldsymbol{b}_i)$ for all non-empty boxes.

The introduced notation is applicable to both the original DETR (Carion et al., 2020) and its more advanced variants such as DAB-DETR (Liu et al., 2022), Deformable DETR (Zhu et al., 2021) as well as any other detector with the end-to-end architecture.

## 3.2. Proposed Critical-Category Objectives

As discussed in Section 1, the overall performance metric evaluated on the validation dataset is not the most effective objective in some real-world scenarios. Category-level fine-grained performance for some specific task-critical categories can be more crucial than the average metrics. Here we provide a practical definition of the critical-category objectives for the detection task, and a corresponding evaluation method when applied to DETR-type detectors.

In classification, class-level performance is often defined as the loss of the model on a subset of the validation dataset that contains objects from a certain group of classes (Tran et al., 2022). However, such definition is not practical for object detection task, as each input image in the dataset contains multiple objects from different categories. Instead, this work defines the critical objective based on the entire validation dataset, but with a *transformed model outputs* and annotations during the loss computation that focus detection towards a certain group of critical object categories.

Formally, assume there are in total $C$ categories in the

dataset. Then, suppose the first $M$ categories are the critical ones for a certain task that requires both an accurate classification and localization. Hence, the rest of categories from $M + 1$ to $C$ are non-critical and a misclassification between them is acceptable. This can be expressed by the transformed prediction $\tilde{\boldsymbol{p}} \in \mathbb{R}^{M+2}$ as

$$\tilde{p}_j = \begin{cases} \hat{p}_j & j = 1 \ldots M, \\ \max \hat{p}_{M+1\ldots C} & j = M + 1, \\ \hat{p}_{C+1} & j = M + 2. \end{cases} \quad (1)$$

The $(M+1)$-th category in $\tilde{\boldsymbol{p}}$ corresponds to "others" which represents non-critical categories. The max function is used to avoid a distinction when classifying non-critical categories. The $(M+2)$-th category in $\tilde{\boldsymbol{p}}$ is used for $\varnothing$ class which is originally defined as the $(C + 1)$-th category in $\hat{\boldsymbol{p}}$.

The same Equation (1) transformation is also applied to the ground truth box label $\boldsymbol{c}$, where all classes $c_{j \in \{M+1,\ldots,C\}}$ are redefined as the $(M + 1)$-th label in the transformed $\tilde{\boldsymbol{c}}$. No change is applied to the ground truth bounding boxes and the predicted boxes as we only define critical performance at the classification granularity to have a simplified yet practical and instructive setting.

The logit transformation can be applied directly to the output of a trained end-to-end model without any change to its architecture or weights $\boldsymbol{\theta}$. The Hungarian matching, loss computation, and mAP computation can be performed without modification as well. We define the loss computed with the original $\hat{\boldsymbol{p}}_i$ and $\boldsymbol{c}_i$ as *"overall objective"* and it is expressed as

$$\mathcal{L}_A(\boldsymbol{\theta}) = \frac{1}{N} \sum_{i=1}^{N} \left( \mathcal{L}_{cls}(\hat{\boldsymbol{p}}_i, \boldsymbol{c}_i) + \mathcal{L}_{box}(\hat{\boldsymbol{b}}_i, \boldsymbol{b}_i) \right). \quad (2)$$

Similarly, our *"critical objective"* is defined with the transformed $\tilde{\boldsymbol{p}}_i$ and $\tilde{\boldsymbol{c}}_i$ as

$$\mathcal{L}_F(\boldsymbol{\theta}) = \frac{1}{N} \sum_{i=1}^{N} \left( \mathcal{L}_{cls}(\tilde{\boldsymbol{p}}_i, \tilde{\boldsymbol{c}}_i) + \mathcal{L}_{box}(\hat{\boldsymbol{b}}_i, \boldsymbol{b}_i) \right). \quad (3)$$

Each objective in Equations (2) and (3) corresponds to either the *"overall performance"* or the *"critical performance"* when evaluating the mAP detection metric with the original or transformed outputs and labels, respectively.

### 3.3. Empirical Evidence of Performance Gaps after Quantization

First, we empirically analyze how quantization affects the critical performance of a DETR model. We apply a symmetric linear quantizer $q(\cdot)$ (Dong et al., 2019) to quantize weights $\boldsymbol{\theta}$ to $Q$ bits of the pretrained DETR checkpoint with ResNet-50 backbone[2], which can be expressed using the

---

[2] https://dl.fbaipublicfiles.com/detr/detr-r50-e632da11.pth

rounding operation $\lfloor \cdot \rceil$ as

$$q(\boldsymbol{\theta}) = \left\lfloor \frac{(2^{Q-1} - 1)\boldsymbol{\theta}}{\max|\boldsymbol{\theta}|} \right\rceil \frac{\max|\boldsymbol{\theta}|}{2^{Q-1} - 1}. \quad (4)$$

We quantize all trainable weights in the DETR model using Equation (4) with an exception of the final feed-forward (FFN) layers for the class and bounding box outputs. Quantization of these FFN layers leads to a catastrophic performance drop in the PTQ setting (Yuan et al., 2022). A 4-bit quantization is applied uniformly to the weights of all layers for all experiments in this empirical study.

Without loss of generality, we define critical categories based on the "super category" labels in the COCO dataset (Lin et al., 2014). In total, 12 super categories are available in the COCO, where each contains from 1 to 10 categories of similar objects. For each selected super category, we consider all the categories within it as critical categories, while the rest of categories as non-critical and transform their logits and labels accordingly. The mAP measured at the transformed output is denoted as the critical mAP of this super category. For example, when measuring the critical performance of "indoor" super category, "book", "clock", "vase", "scissors", "teddy bear", "hair drier", and "toothbrush" are considered as critical categories (the first $M$ categories in the Equation (1) logit-label transformation), while others are set as non-critical. We perform such evaluation for all 12 super categories to understand the category-level impact of DETR quantization.

As shown in Table 1, quantization has a disparate impact on the critical performance of the DETR model. The mAP drop after quantization has an up to 1.7% gap. We further perform 50 epochs of QAT and report the critical performance in Table 2. The performance increases differently for each super category with a gap of up to 1.1% mAP.

## 4. Proposed Methods to Overcome Quantization Gaps

In this section, we theoretically analyze the causes of empirical performance gaps in Section 4.1. Then, we propose our methods to improve such performance from the aspect of quantization scheme design and quantization-aware training objective in Section 4.2 and Section 4.3, respectively.

### 4.1. Causes of Performance Gaps after Quantization

We investigate how quantization affects the critical objective $\mathcal{L}_F(\boldsymbol{\theta})$ for a pretrained DETR model with $\boldsymbol{\theta}$ weights. We obtain the following theoretical results.

**Claim 1: Quantization-induced weight perturbation causes a larger Fisher trace of critical objectives and, therefore, inferior maximum likelihood estimates.** The

Table 1: Super-category mAPs before and after 4-bit uniform quantization, %.

| Super category | Person | Vehicle | Outdoor | Animal | Accessory | Sports | Kitchen | Food | Furniture | Electronic | Appliance | Indoor | Overall |
|---|---|---|---|---|---|---|---|---|---|---|---|---|---|
| Pretrained | 39.4 | 43.9 | 44.4 | 42.5 | 44.6 | 44.2 | 44.8 | 44.7 | 44.7 | 43.8 | 43.9 | 44.9 | 41.9 |
| PTQ 4-bit | 20.1 | 23.3 | 23.9 | 22.3 | 23.9 | 23.7 | 24.2 | 23.9 | 23.7 | 23.4 | 23.5 | 24.0 | 20.9 |
| mAP drop | **19.3** | 20.6 | 20.5 | 20.2 | 20.7 | 20.5 | 20.6 | 20.8 | **21.0** | 20.4 | 20.4 | 20.9 | 21.0 |

Table 2: Super-category mAPs of 4-bit quantized model before and after QAT, %.

| Super category | Person | Vehicle | Outdoor | Animal | Accessory | Sports | Kitchen | Food | Furniture | Electronic | Appliance | Indoor | Overall |
|---|---|---|---|---|---|---|---|---|---|---|---|---|---|
| PTQ 4-bit | 20.1 | 23.3 | 23.9 | 22.3 | 23.9 | 23.7 | 24.2 | 23.9 | 23.7 | 23.4 | 23.5 | 24.0 | 20.9 |
| QAT 4-bit | 34.6 | 38.6 | 39.2 | 37.2 | 39.4 | 38.9 | 39.5 | 39.3 | 39.2 | 38.6 | 38.6 | 39.6 | 36.7 |
| mAP gain | **14.5** | 15.3 | 15.3 | 14.9 | 15.5 | 15.2 | 15.3 | 15.4 | 15.5 | 15.2 | 15.1 | **15.6** | 15.8 |

quantization process replaces the floating-point weights $\boldsymbol{\theta}$ of the pretrained DETR model with the quantized values $q(\boldsymbol{\theta})$ using Equation (4). Effectively, this perturbs the weights away from their optimal values, which leads to an increase in the critical objective value. With the second-order Taylor expansion around $\boldsymbol{\theta}$, the quantization-perturbed loss $\mathcal{L}_F(q(\boldsymbol{\theta}))$ can be approximated using the non-perturbed objective $\mathcal{L}_F(\boldsymbol{\theta})$ as

$$\mathcal{L}_F(q(\boldsymbol{\theta})) \approx \mathcal{L}_F(\boldsymbol{\theta}) + \boldsymbol{g}^T\boldsymbol{\Delta} + \boldsymbol{\Delta}^T\boldsymbol{H}\boldsymbol{\Delta}/2, \quad (5)$$

where the gradient $\boldsymbol{g} = \mathbb{E}\left[\partial\mathcal{L}_F(\boldsymbol{\theta})/\partial\boldsymbol{\theta}\right]$, the Hessian $\boldsymbol{H} = \mathbb{E}\left[\partial^2\mathcal{L}_F(\boldsymbol{\theta})/(\partial\boldsymbol{\theta}\partial\boldsymbol{\theta}^T)\right]$ and the weight perturbation or the quantization error $\boldsymbol{\Delta} = q(\boldsymbol{\theta}) - \boldsymbol{\theta}$.

Assuming the pretrained model converges to a local minimum, the first-order term can be ignored because $\boldsymbol{g} \to 0$ (Le-Cun et al., 1989) and Equation (5) can be rewritten as

$$\mathcal{L}_F(\boldsymbol{\theta}) - \mathcal{L}_F(q(\boldsymbol{\theta})) \propto -\boldsymbol{\Delta}^T\boldsymbol{H}\boldsymbol{\Delta}. \quad (6)$$

For large models such as DETR, computation of the exact Hessian matrix $\boldsymbol{H}$ is practically infeasible. Previous research (Kwon et al., 2022) shows that the Hessian estimate can be derived as the negative of Fisher information matrix $\mathcal{I}$ by

$$\boldsymbol{H} = -\mathcal{I} = -\mathbb{E}\left[\frac{\partial\mathcal{L}_F(\boldsymbol{\theta})}{\partial\boldsymbol{\theta}}\frac{\partial\mathcal{L}_F(\boldsymbol{\theta})}{\partial\boldsymbol{\theta}^T}\right]. \quad (7)$$

Alternatively, we can interpret Equation (7) for a discriminative model $f_{\boldsymbol{\theta}}(\boldsymbol{x})$ from Section 3 that maximizes log-likelihood of the $p(y|\boldsymbol{x}, \boldsymbol{\theta})$ density function using the empirical dataset $\mathcal{D}$ with the loss $\mathcal{L}_F(\boldsymbol{\theta})$ (Gudovskiy et al., 2021) as

$$\begin{aligned} \mathcal{I} &= \mathbb{E}_{\mathcal{D}}\left[\frac{\partial\mathcal{L}_F(\boldsymbol{\theta})}{\partial\boldsymbol{\theta}}\frac{\partial\mathcal{L}_F(\boldsymbol{\theta})}{\partial\boldsymbol{\theta}^T}\right] \\ &= \frac{1}{N}\sum_{i=1}^N\left(\frac{\partial\log p(y_i|\boldsymbol{x}_i)}{\partial\boldsymbol{\theta}}\frac{\partial\log p(y_i|\boldsymbol{x}_i)}{\partial\boldsymbol{\theta}^T}\right). \end{aligned} \quad (8)$$

In practice, we can assume $\mathcal{I}$ to be diagonal (Soen & Sun, 2024), which simplifies Equation (6) to

$$\begin{aligned} \mathcal{L}_F(\boldsymbol{\theta}) - \mathcal{L}_F(q(\boldsymbol{\theta})) &\propto \boldsymbol{\Delta}^T\mathcal{I}\boldsymbol{\Delta} \\ &= \sum_i \boldsymbol{\Delta}_i^2\|\partial\mathcal{L}_F(\boldsymbol{\theta})/\partial\boldsymbol{\theta}_i\|_2^2 = \sum_i \boldsymbol{\Delta}_i^2\mathcal{I}_{ii}, \end{aligned} \quad (9)$$

where the latter result represents a sum of Fisher trace elements $(\text{tr}(\mathcal{I}) = \sum_i \mathcal{I}_{ii})$ weighted by the squared quantization error over each $i$-th element of $\boldsymbol{\theta}$.

Equation (9) provides a feasible yet effective sensitivity metric to estimate the impact of quantization noise. It analytically connects the quantization-induced weight perturbation with the maximum likelihood estimation in Equation (8) for critical objectives using Fisher information framework (Ly et al., 2017). Hence, an objective with larger sensitivity leads to inferior maximum likelihood estimates, i.e. the critical performance.

**Claim 2: Sharp loss landscape leads to a poor test-time generalization for critical categories after quantization-aware training.** During the conventional QAT process, weights of the DETR model are trained to minimize the overall objective $\mathcal{L}_A(q(\boldsymbol{\theta}))$. Nevertheless, a convergence of $\mathcal{L}_A$ does not guarantee good performance on all critical objectives $\mathcal{L}_F$. When compared to the overall objective, the critical objective with the focus on a subset of classes can be quickly minimized by the model during the training process which leads to a tendency of overfitting. The overfitting phenomenon is more severe with a larger model or with more classes in the overall training task.

To better analyze the issue of overfitting, we refer to the previous work on loss landscape sharpness (Foret et al., 2021), which finds a positive correlation between the generalization gap of the objective $\mathcal{L}_F$ and the sharpness $\mathcal{S}$ of the loss landscape around the local minima $q(\boldsymbol{\theta})$ of the QAT. The minima sharpness $\mathcal{S}(q(\boldsymbol{\theta}))$ of the quantized model can be estimated as

$$\mathcal{S}(q(\boldsymbol{\theta})) = \max_{\|\boldsymbol{\epsilon}\|_2 \le \rho} \mathcal{L}_F(q(\boldsymbol{\theta}) + \boldsymbol{\epsilon}) - \mathcal{L}_F(q(\boldsymbol{\theta})), \quad (10)$$

where $\rho > 0$ is a $\ell_2$ norm bound for the worst-case weight perturbation $\boldsymbol{\epsilon}$.

Finding the exact solution to the maximization in Equation (10) can be computationally costly. With the details in Appendix A, we can simplify it as

$$\mathcal{S} \approx \max_{||\boldsymbol{\epsilon}||_2 \leq \rho} \boldsymbol{\epsilon}^T \partial \mathcal{L}_F(q(\boldsymbol{\theta}))/\partial \boldsymbol{\theta} \propto \operatorname{tr}(\mathcal{I}). \qquad (11)$$

Hence, the trace of the diagonal Fisher information matrix approximates the sharpness of the critical loss landscape for the quantized model. Sharp loss landscape leads to inferior test-time critical-category performance after QAT.

### 4.2. Fisher-aware Mixed-Precision Quantization Scheme

With the derived quantization impact on the loss in Equation (9), we propose a mixed-precision quantization scheme that minimizes the quantization effects within a model-size budget, which is defined as

$$\min_{Q_{1:L}} \sum_{i=1}^{L} \boldsymbol{\Delta}_i^2 \left\| \partial \left( \alpha \mathcal{L}_A(\boldsymbol{\theta}) + \mathcal{L}_F(\boldsymbol{\theta}) \right) / \partial \boldsymbol{\theta}_i \right\|_2^2,$$
$$\text{s.t.} \sum_{i=1}^{L} Q_i \left\| \boldsymbol{\theta}_i \right\|_0 \leq B, \qquad (12)$$

where $\boldsymbol{\theta}$ is the weight vector of all $L$ layers in the model, $\boldsymbol{\theta}_i$ is its $i$-th layer subset, and $\boldsymbol{\Delta}_i = q(\boldsymbol{\theta}_i) - \boldsymbol{\theta}_i$ is the $i$-th layer's quantization error when quantized to $Q_i$ bits. The budget $B$ is the model size allowance. The optimization problem in Equation (12) can be efficiently solved as an Integer Linear Programming (ILP) problem (Dong et al., 2020; Yao et al., 2021) with the discrete integer values for quantization precision $Q_i$.

Note that in Equation (12) we employ the Fisher information of both critical $\mathcal{L}_F$ and overall $\mathcal{L}_A$ objectives. This approach achieves good overall performance and increases the critical performance of interest. A hyperparameter $\alpha$ balances $\mathcal{L}_F$ and $\mathcal{L}_A$, which is selected using empirical cross-validation.

### 4.3. Fisher Trace Regularization for Quantization-aware Training

Previous line of work on Sharpness-aware Minimization (SAM) (Foret et al., 2021; Liu et al., 2021) directly optimizes the sharpness estimate from Equation (10) by adding the worst-case weight perturbation in the training. However, we find that the complicated DETR architecture and its objective lead to a poor convergence for SAM-based methods. Moreover, a case with several critical objectives would involve multiple rounds of weight perturbation. Hence, this approach with explicit weight perturbation leads to optimization that is not scalable in our setup.

Instead, to minimize loss sharpness $\mathcal{S}(q(\boldsymbol{\theta}))$ during the DETR QAT optimization, we propose to follow the implicit

sharpness derivation in Equation (11). Specifically, for a critical objective $\mathcal{L}_F$, we add the Fisher trace regularization as

$$\min_{\boldsymbol{\theta}} \mathcal{L}_A(q(\boldsymbol{\theta})) + \lambda \operatorname{tr}(\mathcal{I}_F), \qquad (13)$$

where $\lambda \geq 0$ is the strength of the regularization, and $\mathcal{I}_F$ denotes the Fisher information matrix of the critical objective $\mathcal{L}_F(q(\boldsymbol{\theta}))$ w.r.t. weights $\boldsymbol{\theta}$.

In addition to the DETR training loss terms in Equations (2) and (3), we further add a distillation loss (Hinton et al., 2015) between the quantized (student) model and the pre-trained full-precision (teacher) model to follow a common QAT practice (Dong et al., 2020; Yang et al., 2021). The distillation objective consists of a KL-divergence loss for class logits of the student and teacher models, and a $\ell_1$ loss for the corresponding bounding box coordinates. Since we expect the student model to have the same behavior as the teacher model, the distillation loss uses a fixed one-to-one mapping between the predicted boxes of the two models without performing the Hungarian matching.

## 5. Experiments

### 5.1. Experimental Setup

**Datasets and metrics.** We follow DETR (Carion et al., 2020) setup and use two variants of the COCO 2017 dataset (Lin et al., 2014): COCO detection and COCO panoptic segmentation. The detection dataset contains 118K training images and labels with 80 categories combined into 12 super categories. The panoptic dataset consists of 133K training examples and corresponding labels with 133 categories and 27 super categories. Both variants contain 5K data points in the validation set, which we use to evaluate both the overall and critical mAP in our experiments. Additional CityScapes (Cordts et al., 2016) dataset evaluations are reported in Appendix B.

We follow Section 3.3 and define the critical mean average precision (mAP) for each super category by considering all classes within it as critical while the rest of classes are non-critical. All mAPs reported in the tables are in percentage points. In case of COCO panoptic dataset, we report the box detection $\text{mAP}_{\text{box}}$.

**Model architectures.** We conduct the majority of our experiments on the DETR model with ResNet-50 backbone (DETR-R50). To show the scalability, we also experiment with larger ResNet-101 backbone (DETR-R101) and more advanced architectures such as DAB-DETR (Liu et al., 2022) and Deformable DETR (Zhu et al., 2021).

**Implementation details.** We perform quantization of the pretrained models using their public checkpoints. We apply symmetric layer-wise weight quantization using Equation (4), where weights are scaled by the $\max|\boldsymbol{\theta}|$ without

clamping. We keep normalization and softmax operations at full precision. We compute Fisher trace for our method using all training set for sensitivity analysis. But for implementation of HAWQ-V2 (Dong et al., 2020) baseline, we randomly sample 1,000 training images due to high computational cost of Hessian estimation. We solve mixed-precision quantization problem in Equation (12) by the ILP with 3- to 8-bit budget $B$ for each layer. We perform QAT with the straight-through gradient estimator (Bengio et al., 2013) for 50 epochs with 1e-5 learning rate. Regularization strength $\lambda$ in Equation (13) grows linearly from 1e-3 to 5e-3 throughout the training when our Fisher regularization is applied. In all experiments we report the mean and, if shown, $\pm$ standard error of the final 5 epochs of training to mitigate training variance.

## 5.2. Quantitative Results of Fisher-aware Quantization

We compare the proposed Fisher-aware mixed-precision quantization scheme from Section 4.2 with the linear uniform quantization (Polino et al., 2018) and the mixed-precision HAWQ-V2 (Dong et al., 2020) baselines. Tables 3 and 4 report the critical mAP of super category "person", "animal", and "indoor" for the COCO detection and panoptic segmentation datasets, respectively. We apply the baselines and our Fisher-overall variant for ablation study with only the overall objective $\mathcal{L}_A$, and evaluate quantized models on the selected super categories. Similarly, we report results for the proposed Fisher-critical scheme with the fine-grained $\mathcal{L}_F$ objective.

With the same mixed-precision quantization budget, our Fisher-aware method consistently outperforms uniform quantization and the mixed-precision scheme derived from HAWQ-V2 on different models and datasets. We note that the improvement of the HAWQ-V2 over the uniform quantization is not consistent on DETR-based models. This is caused by the instability of Hessian trace estimation for the complex DETR architecture and the harder object detection task. Fisher-aware approach, on the other hand, is stable. In addition, we compare the time to estimate the Fisher and the Hessian traces for a batch of images on P100 GPU and find that the Fisher trace can be estimated with $200 - 300\times$ *less latency* than the Hessian one. This allows us to estimate Fisher trace with a large amount of training data, which leads to a higher accuracy and stability.

Quantitatively, our Fisher-critical scheme on COCO detection dataset improves critical mAP by up to 0.2% for DETR-R50, 0.5% for DETR-R101, 0.8% for DAB DETR-R50, and 0.4% for Deformable DETR-R50, respectively. With more categories in the panoptic dataset, the impact of quantization on each individual category becomes even higher. Fisher-aware quantization variant with the overall objective only ($\mathcal{L}_A$) improves critical mAP by about $2\times$ over the uniform quantization baseline. Further improvement on critical mAP is consistently achieved with the proposed Fisher-critical quantization scheme that incorporates the fine-grained objective $\mathcal{L}_F$. These results shows the *importance of the proposed scheme with critical objectives* when applying object detection models to real-world applications. Moreover, the common overall mAP metric is not significantly affected when using the conventional $\mathcal{L}_A$ objective or the proposed scheme with $\mathcal{L}_F$ as additionally evaluated in the Appendix B.

## 5.3. Quantitative Results for QAT with Fisher-trace Regularization

Table 5 compares the post-QAT results when using the conventional overall loss $\mathcal{L}_A$ only vs. our approach with Fisher-trace regularization from Section 4.3 on the COCO detection and panoptic datasets. The experimental results show that the proposed regularization further improves critical-category metrics.

When combined with the mixed-precision quantization scheme from Section 4.2, our method on COCO detection dataset (Table 5 (top)) leads to a 1.15% and 0.48% critical ("person" class) performance improvement for DETR-R50 model over the uniform quantization in Table 3 for 4-bit ($34.6\% \rightarrow 35.75\%$) and 6-bit ($37.3\% \rightarrow 37.78\%$) precision, respectively. Note that our regularization scheme has a negligible impact on the overall mAP: $37.07\% \rightarrow 36.97\%$ for 4-bit and $39.67\% \rightarrow 39.70\%$ for 6-bit precision, respectively.

The proposed regularization further increases critical performance on COCO panoptic dataset (Table 5 (bottom)) by 0.11% and 0.34% mAP for, correspondingly, 4-bit and 5-bit precision settings when compared to our PTQ results in Table 4. The uniform PTQ quantization significantly underperforms in this setting.

Ablation study in Appendix C analyzes the impact of regularization strength. In addition, we show that our Fisher-trace regularization scheme that minimizes sharpness of the loss landscape improves model's *test-time generalization*. Particularly, it outperforms a common heuristic approach when the critical objective $\mathcal{L}_F$ is simply added to the overall objective $\mathcal{L}_A$ during quantization-aware training.

## 5.4. Qualitative Results of Fisher-trace Regularization

To further show the effectiveness of the Fisher-trace regularization, we compute the Fisher trace of the critical objective on the quantized DETR model after QAT for the person category. We compare the Fisher trace of models with different quantization and training schemes in Table 6 that is estimated using 10,000 data points randomly sampled from the COCO detection dataset.

Table 3: Critical-category mAP after PTQ on COCO detection dataset, %. Our Fisher-critical scheme with the fine-grained objective surpasses others.

| Model | Quant. scheme | 4-bit | | | 6-bit | | |
|---|---|---|---|---|---|---|---|
| | | Person | Animal | Indoor | Person | Animal | Indoor |
| DETR-R50 | Uniform | 34.6 | 37.2 | 39.6 | 37.3 | 40.0 | 42.4 |
| | HAWQ-V2 | $35.31_{\pm0.1}$ | $37.90_{\pm0.2}$ | $40.20_{\pm0.2}$ | $37.29_{\pm0.0}$ | $40.20_{\pm0.1}$ | $42.60_{\pm0.1}$ |
| | Fisher-overall | $35.35_{\pm0.0}$ | $37.96_{\pm0.2}$ | $40.20_{\pm0.2}$ | $37.58_{\pm0.1}$ | $40.74_{\pm0.1}$ | $43.10_{\pm0.1}$ |
| | Fisher-critical | $\mathbf{35.56}_{\pm0.1}$ | $\mathbf{38.10}_{\pm0.1}$ | $\mathbf{40.33}_{\pm0.0}$ | $\mathbf{37.73}_{\pm0.0}$ | $\mathbf{40.86}_{\pm0.1}$ | $\mathbf{43.26}_{\pm0.1}$ |
| DETR-R101 | Fisher-overall | 36.36 | **39.30** | 41.70 | 39.1 | 42.0 | 44.4 |
| | Fisher-critical | **36.42** | 39.23 | **41.80** | **39.2** | **42.5** | **44.9** |
| DAB DETR-R50 | Uniform | 22.32 | 25.68 | 27.60 | 26.24 | 29.76 | 31.88 |
| | HAWQ-V2 | 8.26 | 11.66 | 12.80 | 19.10 | 19.90 | 21.60 |
| | Fisher-overall | 22.82 | 27.02 | **28.96** | 26.06 | 29.20 | 31.32 |
| | Fisher-critical | **23.18** | **27.86** | 27.98 | **26.38** | 29.28 | **31.88** |
| Deformable DETR-R50 | Uniform | 28.9 | 32.8 | 34.3 | 46.0 | 49.1 | 51.4 |
| | Fisher-overall | 42.7 | 46.2 | 48.4 | 46.3 | **49.5** | 51.8 |
| | Fisher-critical | **43.1** | **46.3** | **48.8** | **46.6** | **49.5** | **52.0** |

Table 4: Critical-category mAP$_{box}$ for various post-training quantization (PTQ) schemes on COCO panoptic dataset, %. Our Fisher-critical scheme exceeds others.

| Model | Quant. scheme | 4-bit | | | 5-bit | | |
|---|---|---|---|---|---|---|---|
| | | Person | Animal | Indoor | Person | Animal | Indoor |
| DETR-R50 | Uniform | 8.5 | 11.4 | 12.4 | 8.9 | 13.7 | 16.0 |
| | Fisher-overall | 16.64 | 21.60 | 23.80 | 18.79 | 24.00 | 26.70 |
| | Fisher-critical | **16.68** | **21.69** | **23.85** | **19.05** | **24.15** | **26.87** |

Table 5: Quantization-aware training (QAT) results of the overall performance and for the "person" critical category using the DETR-R50 model on COCO detection (mAP) and panoptic (mAP$_{box}$) datasets, %. Fisher-critical quantization is applied before QAT.

| Model | QAT objective | 4-bit | | 6-bit det. / 5-bit panoptic | |
|---|---|---|---|---|---|
| | | Overall | Person | Overall | Person |
| DETR-R50 (detection) | Overall | $\mathbf{37.07}_{\pm0.07}$ | $35.56_{\pm0.08}$ | $39.67_{\pm0.10}$ | $37.73_{\pm0.02}$ |
| | Fisher reg. | $36.97_{\pm0.06}$ | $\mathbf{35.75}_{\pm0.04}$ | $\mathbf{39.70}_{\pm0.08}$ | $\mathbf{37.78}_{\pm0.01}$ |
| DETR-R50 (panoptic) | Overall | $33.24_{\pm0.10}$ | $16.68_{\pm0.01}$ | $36.08_{\pm0.07}$ | $19.05_{\pm0.09}$ |
| | Fisher reg. | $\mathbf{33.29}_{\pm0.05}$ | $\mathbf{16.79}_{\pm0.03}$ | $\mathbf{36.12}_{\pm0.06}$ | $\mathbf{19.39}_{\pm0.11}$ |

Table 6: Fisher trace of the critical objective when applied to DETR-R50 on COCO detection dataset. In this setting, the person category is considered as critical.

| Precision | Quant. scheme | Regularization | Fisher trace |
|---|---|---|---|
| 4-bit | Uniform | No | 37.3K |
| | Fisher-critical | No | 30.4K |
| | Fisher-critical | Yes | **14.9K** |
| 6-bit | Uniform | No | 88.9K |
| | Fisher-critical | No | 18.2K |
| | Fisher-critical | Yes | **15.5K** |

Both the 4-bit and 6-bit uniform quantization settings lead to the largest Fisher trace on the critical objective, while our Fisher-aware quantization scheme helps to reduce the trace after QAT. Furthermore, the proposed regularization results in the lowest value. This observation confirms our analytical result in Section 4.1, where large Fisher trace indicates the least sharp local minima and, therefore, leads to inferior test-time generalization for critical categories.

## 6. Conclusions

This work investigated the impact of quantization on the fine-grained performance of DETR-based object detectors. Motivated by safety concerns in practical applications, we formulated the critical-category objectives via the logit-label

transformation of the corresponding categories. We empirically found that both the conventional PTQ and QAT cause disparate quantization effects.

We theoretically linked the disparate quantization effects with the sensitivity to the quantization weight perturbation and the sharpness of the loss landscape in the QAT. We characterized both derivations using the trace of the Fisher information matrix w.r.t. model weights. We proposed the Fisher-aware mixed-precision quantization scheme and Fisher-trace regularization to improve the critical-category performance of interest. We hope this work motivates future explorations on the fine-grained impacts of other compression methods in the computer vision area and a general machine learning research.

## Acknowledgement

We thank Panasonic and Berkeley Deep Drive for supporting this research.

## Impact Statement

This paper presents work whose goal is to advance the field of Machine Learning, specifically improving the fine-grained performance of the critical categories of interest of the DETR model. In practical applications, the model's capability of correctly detecting each object category is not equally valuable. Some critical category appears to be more impactful to the general utility of the model, or leads to a more significant impact on the safety and trustworthiness of the application. We hope this work motivates future explorations on the fine-grained impacts of other compression methods in the computer vision area and a general machine learning research.

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

We provide supplementary materials in the following appendices. Specifically, Appendix A provides detailed derivation of Equation (11). Appendix B further analyzes how the overall and critical-category objectives from Equation (12) impact the overall performance metric, and contains additional CityScapes experimental results. Appendix C conducts an ablation study on the impact of regularization strength $\lambda$ for our Fisher-trace regularization and verifies the design choice of using our regularizer instead of a simple summation of overall and critical-category objectives for the QAT optimization. Appendix D provides additional qualitative visualizations of the layer-wise Fisher-aware sensitivity and the corresponding quantization assignments derived using the proposed method.

## A. Detailed Derivation of Equation (11)

The first-order Taylor expansion of the perturbed loss $\mathcal{L}_F(q(\boldsymbol{\theta}) + \boldsymbol{\epsilon})$ in Equation (10) is

$$\mathcal{L}_F(q(\boldsymbol{\theta}) + \boldsymbol{\epsilon}) \approx \mathcal{L}_F(q(\boldsymbol{\theta})) + \boldsymbol{\epsilon}^T \partial \mathcal{L}_F(q(\boldsymbol{\theta}))/\partial \boldsymbol{\theta}. \tag{14}$$

By substituting Equation (14) into Equation (10), the maximization can be simplified as

$$\mathcal{S}(q(\boldsymbol{\theta})) = \max_{\|\boldsymbol{\epsilon}\|_2 \leq \rho} \mathcal{L}_F(q(\boldsymbol{\theta}) + \boldsymbol{\epsilon}) - \mathcal{L}_F(q(\boldsymbol{\theta})) \approx \max_{\|\boldsymbol{\epsilon}\|_2 \leq \rho} \boldsymbol{\epsilon}^T \partial \mathcal{L}_F(q(\boldsymbol{\theta}))/\partial \boldsymbol{\theta}. \tag{15}$$

Note that both the $\boldsymbol{\epsilon}$ and $\partial \mathcal{L}_F(q(\boldsymbol{\theta}))/\partial \boldsymbol{\theta}$ are vectors with the same dimensions as weight vector $\boldsymbol{\theta}$. Then, their inner product achieves the maximum when they are parallel. Therefore, we can solve the maximization in Equation (15) as

$$\begin{aligned}
\mathcal{S}(q(\boldsymbol{\theta})) &\approx \max_{\|\boldsymbol{\epsilon}\|_2 \leq \rho} \boldsymbol{\epsilon}^T \partial \mathcal{L}_F(q(\boldsymbol{\theta}))/\partial \boldsymbol{\theta} \\
&= \frac{\rho}{\|\partial \mathcal{L}_F(q(\boldsymbol{\theta}))/\partial \boldsymbol{\theta}\|_2} \frac{\partial \mathcal{L}_F(q(\boldsymbol{\theta}))}{\partial \boldsymbol{\theta}^T} \frac{\partial \mathcal{L}_F(q(\boldsymbol{\theta}))}{\partial \boldsymbol{\theta}} \\
&\propto \frac{\partial \mathcal{L}_F(q(\boldsymbol{\theta}))}{\partial \boldsymbol{\theta}^T} \frac{\partial \mathcal{L}_F(q(\boldsymbol{\theta}))}{\partial \boldsymbol{\theta}} = \mathrm{tr}(\mathcal{I}),
\end{aligned} \tag{16}$$

which is the final approximation of the loss landscape sharpness in Equation (11).

## B. Ablation Study on Fisher-aware Quantization

Table 3 contains only the critical-category metrics. Here we report the overall mAPs in Tables 7 and 8 to show the impact on the overall performance. In general, Fisher-critical quantization scheme leads to comparable overall metrics with the Fisher-overall scheme, and they both are significantly higher than the conventional uniform and HAWQ-V2 quantization schemes. In some cases, the improvement of critical-category metrics with the proposed scheme also improves the overall performance. This indicates that the addition of such critical-category objective in the sensitivity analysis can be useful for increasing the overall performance as well. This is an interesting direction for future work.

Table 7: Overall mAP on COCO detection dataset for 4-bit precision budget, %.

| Model | Uniform | Fisher-overall | Fisher-person | Fisher-animal | Fisher-indoor |
|---|---|---|---|---|---|
| DETR-R50 | 36.7 | $\mathbf{37.12}_{\pm 0.1}$ | $37.1_{\pm 0.1}$ | $37.0_{\pm 0.1}$ | $36.99_{\pm 0.1}$ |
| DETR-R101 | 37.4 | $\mathbf{38.26}$ | 38.22 | 37.97 | 38.24 |
| DAB DETR-R50 | 22.7 | 24.42 | 25.28 | $\mathbf{25.84}$ | 24.08 |
| Deformable DETR-R50 | 28.8 | 44.1 | $\mathbf{44.5}$ | 44.1 | $\mathbf{44.5}$ |

Table 8: Overall mAP on COCO detection dataset for 6-bit precision budget, %.

| Model | Uniform | Fisher-overall | Fisher-person | Fisher-animal | Fisher-indoor |
|---|---|---|---|---|---|
| DETR-R50 | 39.4 | $39.57_{\pm 0.1}$ | $\mathbf{39.67}_{\pm 0.1}$ | $39.60_{\pm 0.0}$ | $39.61_{\pm 0.1}$ |
| DETR-R101 | 39.2 | 41.8 | 41.8 | $\mathbf{42.1}$ | $\mathbf{42.1}$ |
| DAB DETR-R50 | 28.00 | 27.20 | $\mathbf{28.42}$ | 27.30 | 27.94 |
| Deformable DETR-R50 | 47.8 | 48.1 | $\mathbf{48.5}$ | 48.1 | $\mathbf{48.5}$ |

Finally, we provide additional results of the Fisher-aware quantization scheme with the CityScapes dataset in Table 9. We perform object detection task with DETR model on the CityScapes dataset following the settings of (Wang et al., 2022)[3]. We can observe the same trend that the proposed Fisher-overall scheme significantly outperforms uniform quantization, whereas Fisher-critical scheme further improves the performance of the corresponding critical categories.

Table 9: Critical-category mAP on CityScapes for DETR-R50, %.

| Precision | Quant. scheme | Overall | Critical category | | | |
| | | | Construct | Object | Human | Vehicle |
|---|---|---|---|---|---|---|
| FP | - | 11.7 | 8.7 | 17.8 | 18.0 | 19.0 |
| 4-bit | Uniform | 5.2 | 3.6 | 8.7 | 8.8 | 9.2 |
| | Fisher-overall | 8.8 | 5.5 | 12.6 | 13.8 | 14.6 |
| | Fisher-critical | **9.0** | **6.5** | **13.7** | **14.0** | **14.7** |

## C. Ablation Study on Fisher-trace Regularization

We start with the discussion about the impact of regularization strength $\lambda$ on the overall and the critical performance in the QAT. Similarly to previous work on the regularized training (Yang et al., 2022), $\lambda$ controls the tradeoff between the overall performance and the generalization gap for the critical objective. Table 10 shows the overall and critical mAP during training if we set $\lambda$ to a smaller value e.g., 1e-3. It can be seen that the Fisher trace regularization significantly improves critical mAP during epoch range from 20 to 30 (up to 0.5%). As the training progresses towards convergence, the critical-category performance drops while the overall performance increases which indicating the occurrence of overfitting.

However, setting the $\lambda$ too large (e.g., 5e-3) during the initial epochs of the QAT process significantly affects the convergence of the overall training objective. These observations indicate that during the QAT process, a smaller regularization is needed initially to facilitate convergence, while a larger regularization is needed towards the end to prevent the overfitting. To address this in our work we utilize a linear scheduling of the regularization strength as discussed in Section 5.1, which can be formulated as $\lambda = \max[\lambda_0, \lambda_T t/T]$, where $t$ is the current epoch, $T$ is the total number of epochs, and $\lambda_0, \lambda_T$ are the initial and final regularization strengths, respectively. This scheme leads to higher results in Tables 4 and 5.

Finally, we verify the necessity of applying Fisher-trace regularization during QAT. Specifically, we compare to a common heuristic approach when the critical objective $\mathcal{L}_F$ is simply added to the overall objective $\mathcal{L}_A$. For the Fisher-trace regularization, the motivation comes from our Claim 2 in Section 4.1, where the QAT gap is caused by the *sharp loss landscape*, which leads to a *poor generalization*. Table 11 results confirm that the test-time generalization cannot be improved by the addition of critical objectives to the training loss.

## D. Qualitative Results of Fisher-aware Quantization

### D.1. Fisher-aware Sensitivity vs. Quantization Assignments

We illustrate the Fisher-aware sensitivity and the corresponding quantization assignments for a Fisher-overall scheme from Table 4 when applied to the DETR-R50 model on the COCO panoptic dataset in Figure 2. As shown in the visualization, the backbone layers demonstrate a relatively stable sensitivity distribution, while the transformer encoder and decoder layers show sensitivity distribution with high variance. This is expected given the different functionalities of transformer layers within an attention block (Carion et al., 2020). Then, the quantization assignments are performed with the clear correlation between the sensitivity magnitude for each layer and the budget constraints when solving the ILP from Equation (12). Additionally, we visualize DETR-R50, DETR-R101, DAB DETR-R50, and Deformable DETR-R50 models on the COCO detection dataset in Figure 3, and, additionally, draw DETR-R101 on the COCO panoptic dataset in Figure 4.

---

[3]https://github.com/encounter1997/DE-DETRs

Table 10: QAT performance of DETR-R50 on COCO detection dataset. The person category is considered as critical and 4-bit Fisher-critical quantization scheme is applied. The mAP metrics at each epoch are reported using overall/critical format, %.

| $\lambda$ | Epoch 10 | Epoch 20 | Epoch 30 | Epoch 40 | Epoch 50 |
|---|---|---|---|---|---|
| 0 | 36.8/34.8 | 36.7/34.7 | 36.9/34.9 | 37.3/35.1 | 37.2/35.2 |
| 1e-3 | 36.5/34.5 | 36.9/**35.2** | 36.8/**35.3** | 37.1/35.0 | 37.2/35.1 |

Table 11: QAT performance of DETR-R50 model on COCO detection (left) and panoptic (right) datasets. All models are quantized using the Fisher-critical scheme with 4-bit budget for detection and 5-bit budget for panoptic dataset, respectively.

| QAT objective | Overall | Person | QAT objective | Overall | Person |
|---|---|---|---|---|---|
| overall | $37.07_{\pm 0.07}$ | $35.56_{\pm 0.08}$ | overall | $36.08_{\pm 0.07}$ | $19.05_{\pm 0.09}$ |
| overall+critical | $37.11_{\pm 0.04}$ | $35.39_{\pm 0.06}$ | overall+critical | $36.07_{\pm 0.05}$ | $19.19_{\pm 0.10}$ |
| Fisher reg. | $36.97_{\pm 0.06}$ | $\mathbf{35.75}_{\pm 0.04}$ | Fisher reg. | $\mathbf{36.12}_{\pm 0.06}$ | $\mathbf{19.39}_{\pm 0.11}$ |

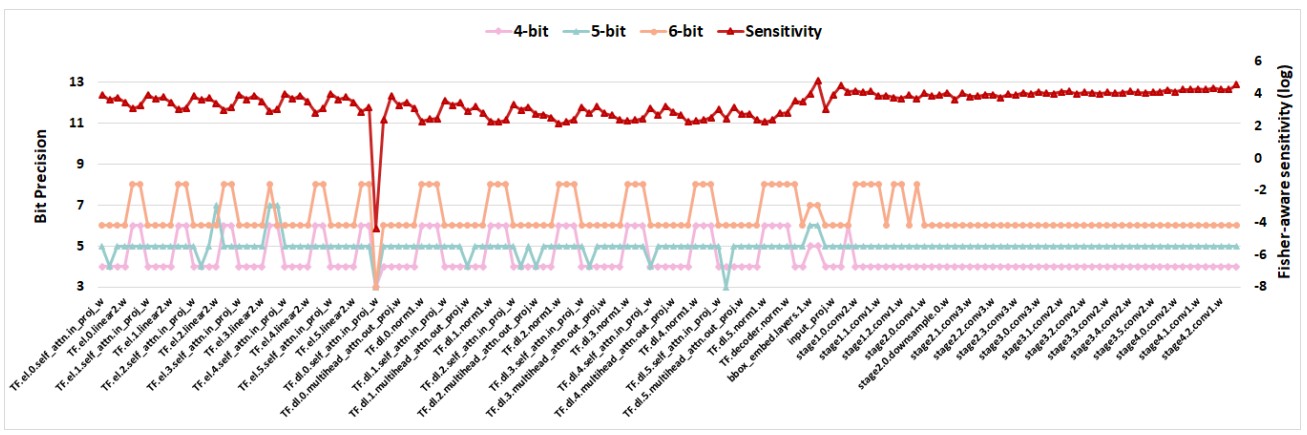

Figure 2: Bit precision vs. layer-wise sensitivity for DETR-R50 on COCO panoptic dataset. There is a clear correlation between the number of bits and our sensitivity metric.

## D.2. Overall vs. Critical-category Objectives for Quantization

Next, we visually compare mixed-precision quantization assignments in our Fisher-aware scheme when the conventional "overall" objective or the proposed "critical-category" objective are employed. Figure 5 compares the assignments for the DETR-R50 model on COCO detection dataset when applied to the person category that corresponds to the quantitative result in Table 3 (top). As shown in the figure, the inclusion of critical objective into the ILP leads to a *significant change in the precision assigned to certain layers*. In particular, our Fisher-critical scheme has more peaks and lows than a more smooth conventional scheme. This illustrates high sensitivity of layers to the critical-category objective. By adding the objectives of interest, it is possible to *improve model's quantization at the fine-grained level*. Additionally, Figures 6 to 8 compare the assignments for different models and critical categories reported in Tables 3 and 4.

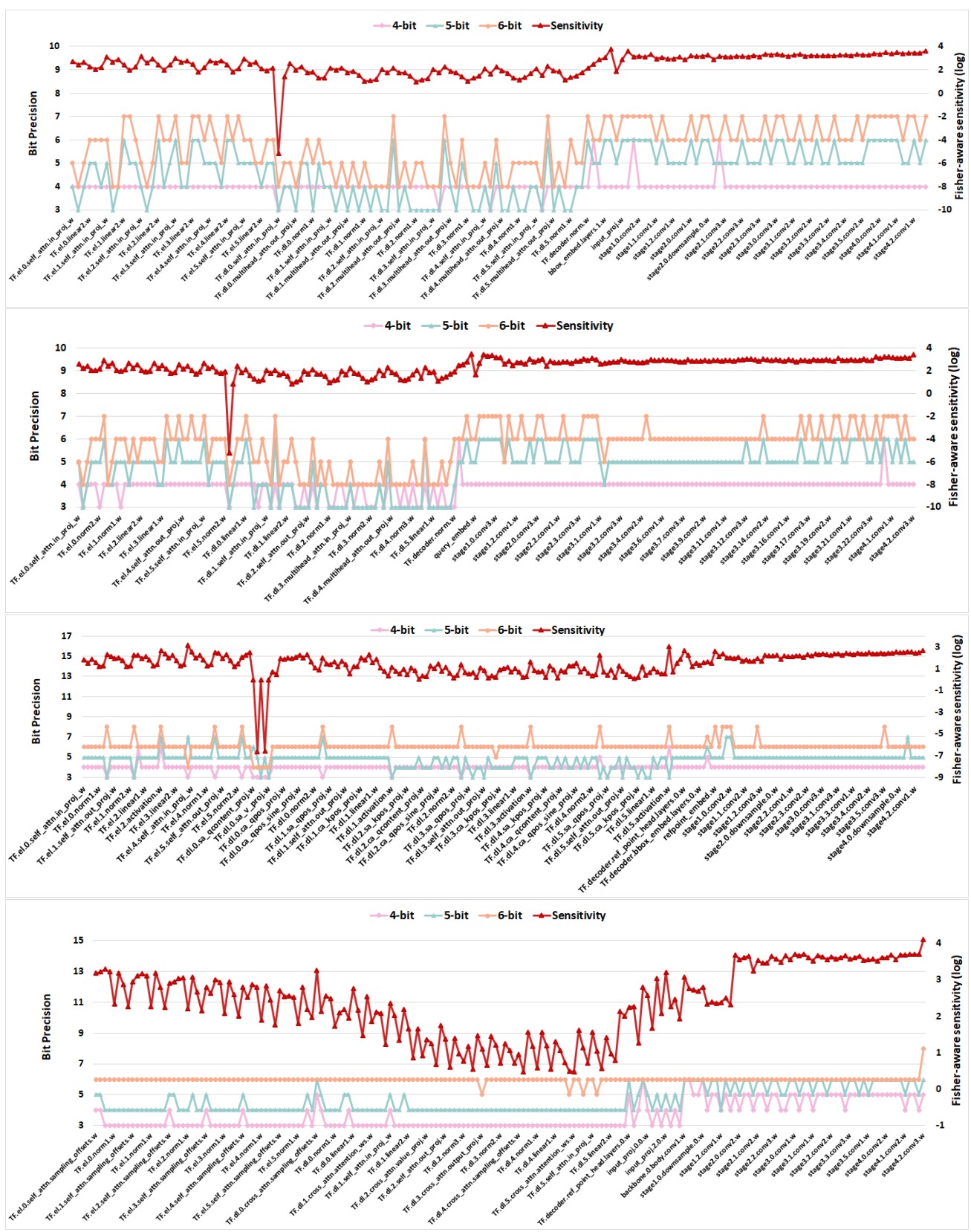

Figure 3: Bit precision vs. layer-wise sensitivity for DETR-R50, DETR-R101, DAB DETR-R50 and Deformable DETR-R50 on COCO detection dataset, respectively.

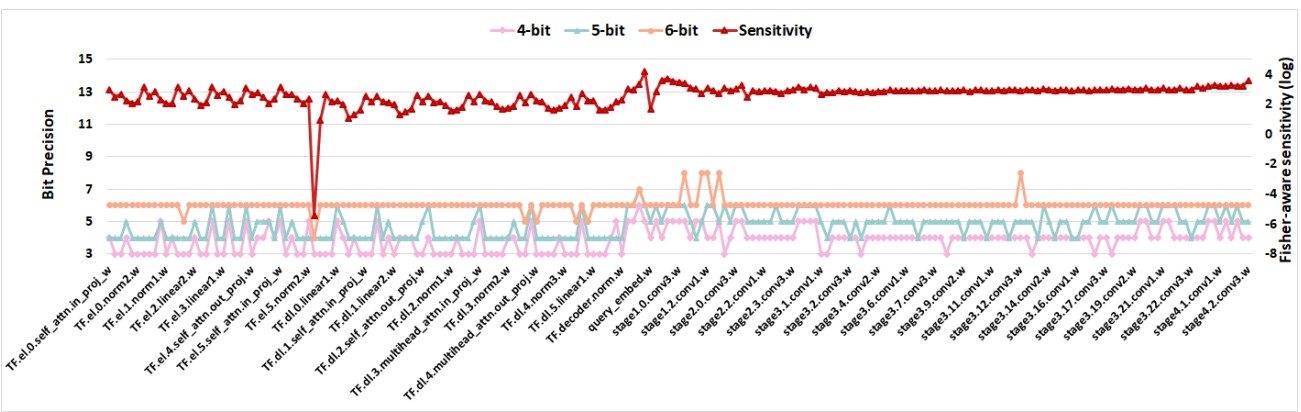

Figure 4: Bit precision vs. layer-wise sensitivity for DETR-R101 on COCO panoptic dataset.

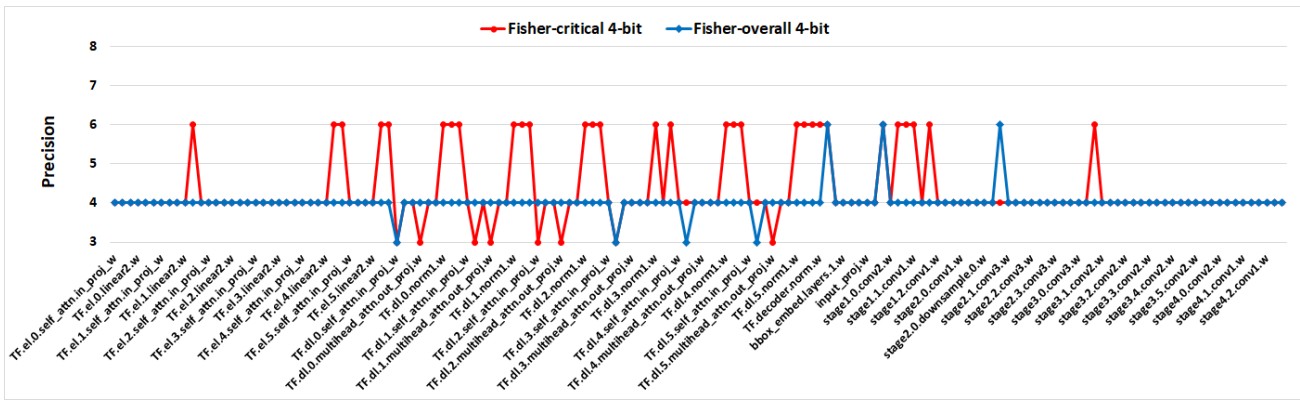

Figure 5: Comparison of Fisher-critical and Fisher-overall assignments for DETR-R50 on COCO detection dataset when applied to the person category. Our critical objective leads to a significant change in the precision assigned to detector's layers.

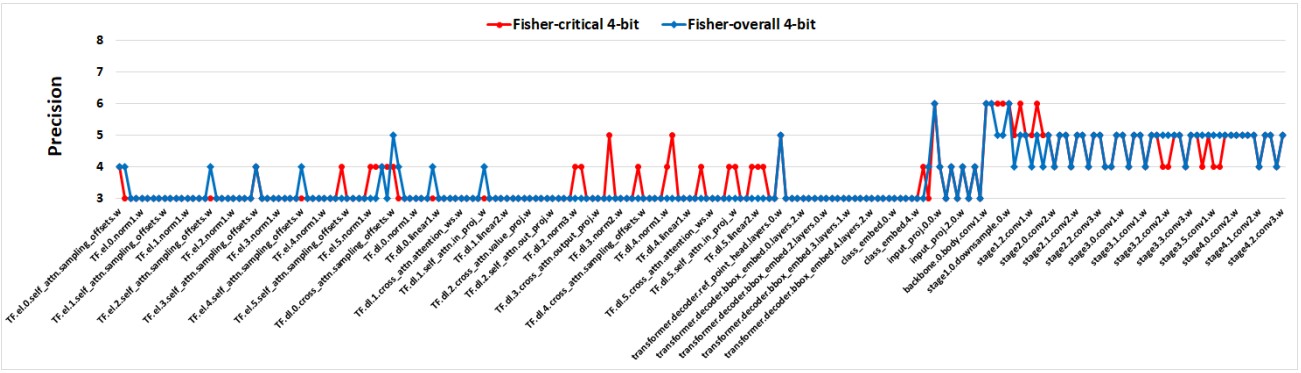

Figure 6: Comparison of Fisher-critical and Fisher-overall assignments for Deformable DETR-R50 on COCO detection dataset when applied to **person category**.

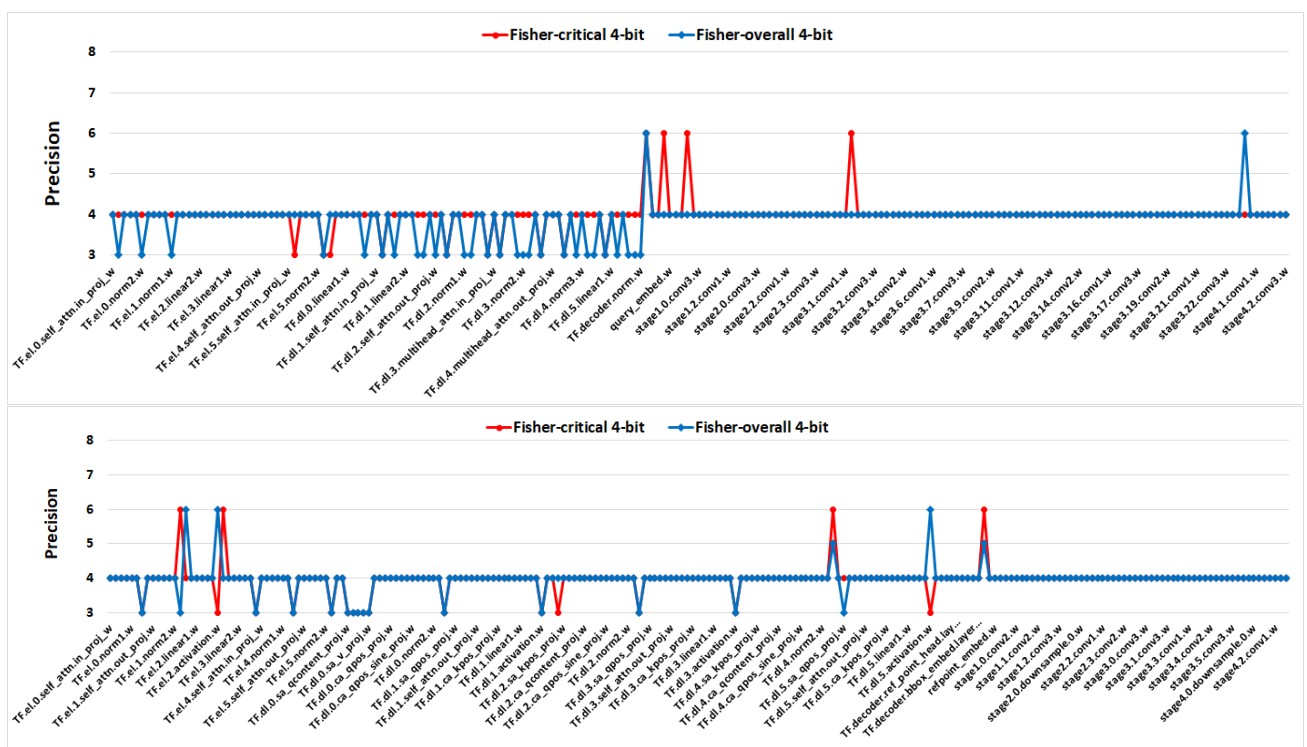

Figure 7: Comparison of Fisher-critical and Fisher-overall assignments for DETR-R101 and DAB DETR-R50 on COCO detection dataset when applied to **indoor** and **animal** categories, respectively.

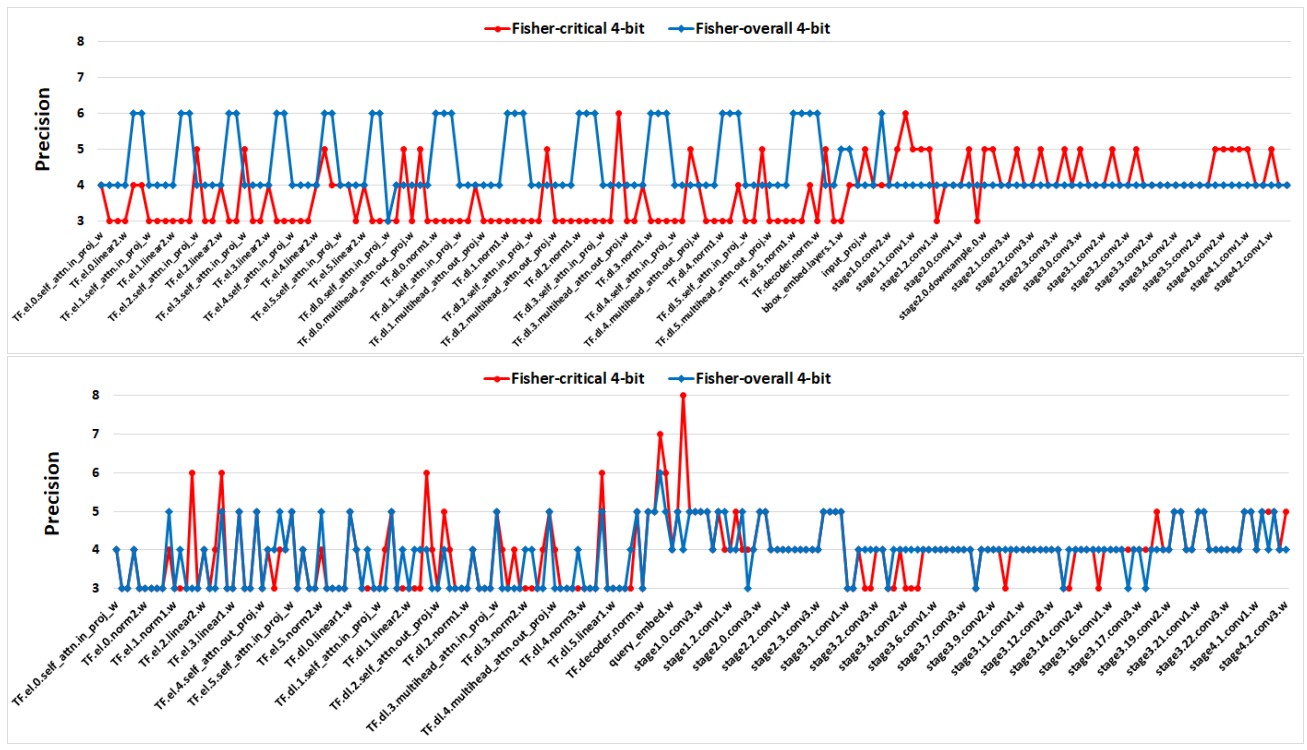

Figure 8: Comparison of Fisher-critical and Fisher-overall assignments for DETR-R50 and DETR-R101 on COCO panoptic dataset when applied to **animal** and **person** categories, respectively.

