# OpenReview forum: "Fisher-aware Quantization for DETR Detectors with Critical-category Objectives"
_ICML.cc/2024/Workshop/WANT — WANT@ICML 2024 Poster_

### Official Review · Reviewer_nmpL · 2024-06-13
**Interesting Fisher-information based quantization method for detection transformer**

**Confidence:** 4

**Summary:**

The authors consider the quantization of detection transformers (DETR) for object detection tasks. They first introduce a concept of critical category for the classification objective. Then, they study the impact of quantization on prediction performances for these categories. They later analyze this impact with Fisher information as an approximation of the Hessian. From this analysis, they propose a Fisher-aware quantization technique. Experiments show that this technique leads to improved mAP for DETR models trained on COCO datasets compared to uniform and hessian aware (HAWQ-V2) approaches to quantization.

The paper presents a method for the efficient training of transformers in computer vision. It is thus aligned with the topics of the workshop. I suggest to accept the paper.

**Strengths:**

- The paper is clear, well organized and easy to follow;
- The idea is simple and mathematically motivated;
- Experiments show consistent improvement of the mAP with their method compared to Uniform and HAWQ-V2.

**Weaknesses:**

The paper introduces critical-category, which are well motivated in the application of autonomous driving they mention. But how well does it transpose to other applications? For instance, what justifies the choice for _Person_, _Animal_, and _Indoor_ in their experiments?

While the Fisher-based quantization technique is being presented in the context of critical-category, the connection between the two concepts could be emphasized. Is the Fisher-based quantization also interesting because it can address the case of critical-category, while other techniques like HAWQ-V2 can only operate at the level of all categories?

It would be interesting to compare the impact of the different quantization in a quantization-aware training on additional metrics like the latency. Especially, when Fisher information is easier to obtain than the full Hessian.

---

### Official Review · Reviewer_jV7J · 2024-06-15
**Studying the effects of quantization and category importance in object detection**

**Confidence:** 4

**Summary:**

For efficiency, NN are often quantized. The article claims that category importance should play an important role in avoiding performance loss in object detection methods. They propose the use of the Fisher information matrix as a proxy for the Hessian to find a better objective. They also advocate using the trace of the FIM to estimate the sharpness of the loss landscape and utilise this information as a regulariser.

They show convincing improvement in mAP over the detection of critical classes in their results.

**Strengths:**

The paper is interesting and on-topic for the conference. It makes a few interesting points and shows good results when critical categories are important.

**Weaknesses:**

The paper is complex and while well-written not very easy to follow.  The appendix is actually important to read to better understand the article.

**Limitations:**

I don't think this is true that the importance of critical fine-grained objectives is really overlooked in the literature, and I'm not sure that such objectives should be actually quantized to any large degree in real-world applications (down to 4 bits?). The mAP numbers showed in the results are actually very low. While the proposed methodology helps matters significantly, it does not make it actually exploitable.

**Suggestions:**

The paper is going to be harder to understand if the appendix is not also published. I appreciated the more mathematically-inclined lean of the paper, so unfortunately I'm not quite sure what to suggest as improvement for readability. Perhaps try to gain some room by shortening page 2 of the paper. I didn't find figure 1 actually helpful.

Explain why the MILP formulation of eq. (12) can be efficiently solved. A MILP is typically NP-hard, which is the opposite of efficient. Is the problem always small ? is there a specific method that can be used in this case ?

---

### Meta-Review · Area_Chair_VzMA · 2024-06-17

**Recommendation:** Accept (Poster)
**Confidence:** 4

**Metareview:**

The paper proposes a new detection model that combines DETR-type architectures with low (4) bit quantization. The reviewers agree that the paper is interesting, with novel elements. The AC agrees and recommends for acceptance. Please try to address the feedback received by the camera ready.

---

### Decision · Program_Chairs · 2024-06-17

**Decision:**

Accept (Poster)

**Comment:**

We thank the authors for their time and contribution to WANT and we are pleased to share that after the reviewing process the paper has been accepted. Congratulations! We encourage the authors to consider reviewers' feedback for the improvement of the camera-ready version. We hope to see you in person at the workshop and brainstorm on efficient training research together!